# Energy Schedule Setting Based on Clustering Algorithm and Pattern Recognition for Non-Residential Buildings Electricity Energy Consumption

**Yu Cui, Zishang Zhu \*, Xudong Zhao \***  **and Zhaomeng Li**

Center for Sustainable Energy Technologies, Energy and Environment Institute, University of Hull,
Hull HU6 7RX, UK; y.cui-2018@hull.ac.uk (Y.C.); lzmeng@mail.ustc.edu.cn (Z.L.)
\* Correspondence: zishang.zhu@hull.ac.uk (Z.Z.); xudong.zhao@hull.ac.uk (X.Z.)

**Abstract:** Building energy modelling (BEM) is crucial for achieving energy conservation in buildings, but occupant energy-related behaviour is often oversimplified in traditional engineering simulation methods and thus causes a significant deviation between energy prediction and actual consumption. Moreover, the conventional fixed schedule-setting method is not applicable to the recently developed data-driven BEM which requires a more flexible and data-related multi-timescales schedule-setting method to boost its performance. In this paper, a data-based schedule setting method is developed by applying K-medoid clustering with Principal Component Analysis (PCA) dimensional reduction and Dynamic Time Warping (DTW) distance measurement to a comprehensive building energy historical dataset, partitioning the data into three different time scales to explore energy usage profile patterns. The Year–Month data were partitioned into two clusters; the Week–Day data were partitioned into three clusters; the Day–Hour data were partitioned into two clusters, and the schedule-setting matrix was developed based on the clustering result. We have compared the performance of the proposed data-driven schedule-setting matrix with default settings and calendar data using a single-layer neural network (NN) model. The findings show that for the data-driven predictive BEM, the clustering results-based data-driven schedule setting performs significantly better than the conventional fixed schedule setting (with a 25.7% improvement) and is more advantageous than the calendar data (with a 9.2% improvement). In conclusion, this study demonstrates that a data-related multi-timescales schedule matrix setting method based on cluster results of building energy profiles can be more suitable for data-driven BEM establishment and can improve the data-driven BEMs performance.

**Keywords:** energy schedule; occupation behavior; k-medoids clustering; Dynamic Time Warping distance

## 1. Introduction

Occupant energy behaviour is currently acknowledged as a significant impact factor in building energy simulation, contributing to 4% to 30% of energy savings [1]. However, modelling occupant energy-related schedules is difficult due to their stochastic and complex nature [2], making it challenging to define realistic schedules. In conventional building energy simulation software (i.e., EnergyPlus and DOE), occupant energy-related behaviour is simulated using fixed schedules, extracted from standards such as ASHRAE [3] or national survey data [4]. However, this approach lacks flexibility and regional differences are not considered. Consequently, it brings high uncertainty to prediction models and results in a large deviation between predicted and actual energy usage [5].

Customization is another common idea for developing schedules, based on detailed survey and sensing data. However, due to the difficulty in data collection and the high working load, its application is very limited.

Recent developments in building smart electricity meters have resulted in vast amounts of historical energy data, enabling the application of algorithms. Among them, clustering

algorithms are attractive and widely used in building benchmarks, knowledge discovery and fault diagnosis. For example, k-means clustering was employed to conduct building energy patterns discovering and recognition [6], and subtractive clustering combined with ANN was applied to HVAC system error detection and diagnosis [7].

Additionally, many studies have proposed strategies for appropriately defining building energy schedules based on historical energy data and computational algorithms [8]. The popular method includes the data mining approach, probabilistic approach [9], and agent-based approach [10,11].

For the data mining approach, there are two main kinds of technologies that could be integrated with BEM analysis. The first one is the traditional global optimization approach, which is also known as the exact approach. For example, an 'Energy-use per person (K)' model with Multiple non-linear regression (MNLR) and Deep neural network (DNN) has been developed to conduct occupancy analysis and further estimate the energy saving potential, which is a supervision and global optimization model [12]. Using the exact approach could guarantee the global optimization solution (minimum distance) is found in a statistical way. Still, it often lacks computational efficiency and is hard for high-dimensional data. With algorithm development, a high performance and better applicability exact approach (such as SOS-SDP: An Exact Solver for Minimum Sum-of-Squares Clustering algorithm [13]) will provide new opportunities to integrate with BEM. Another one is the unsupervised heuristic clustering approach which is a more commonly used technology because the energy pattern analysis is a computational complexity problem with high dimensional data and a need for interpretability. However, although the heuristic approach has good applicability and interpretability, there is often a concern regarding solution quality. Unlike the exact approach, the heuristic approach is very sensitive to the selection of initial points, and it might stop at a local minimize point. Therefore, it requires careful approach to choosing and applying the heuristic rule and further determining the initial clustering medoids. The commonly used technologies in this category included k-means [14,15], k-shape clustering [16,17] and k-medoid clustering [18].

Currently, many studies have attempted to set up data-driven schedules to integrate with engineering building simulation models to improve their accuracy [15,19]. These efforts have not fully demonstrated the advantages of data-driven schedule setting. On the other hand, data-based BEM also lacks applicable schedule settings [20,21], and the current solution of using calendar data with set points value to represent the time series is less reasonable. Therefore, combining a data-related schedule setting with a data-driven BEM is a better choice. Compared to conventional fixed schedules, data-driven schedules are more flexible and take into account all aspects of building energy usage. However, they are not easy to explain or adjust. Data-driven models are different from engineering models, as they prioritize speed and accuracy over physical compliance and being explainable. With the data-driven prediction model, schedule setting can eliminate the consideration of real occupant behaviour and interaction, prescribing energy usage patterns solely based on energy profile patterns extracted from the clustering method.

Furthermore, research on energy profile pattern recognition and schedule development has focused on conducting in-depth research on a single time-scale and lacks comprehensive research on multiple time-scale clustering and schedule matrix development [17]. Some research indicates the schedule setting based on the minimum temporal resolution has the greatest impact on the performance of the energy model [22], but schedules on larger timescales are also important for a more holistic measure of energy consumption patterns. Developing a schedule-setting matrix rather than a vector is more reasonable.

By reviewing current studies, there is a lack of an elaborate schedule-setting method for the data-driven BEM which should be more flexible and data-related than the fixed schedule setting and more representative than the calendar data. To address the pre-mentioned issues, this paper aims to design a multi-timescales schedule-setting matrix based on clustering results from building energy profiles in non-residential buildings, which is more suitable for data-driven BEM. This paper systematically clusters a comprehensive

building energy historical dataset into three different time scales (Year–Month, Week–Day, and Day–Hour) to explore the energy usage profile patterns. K-medoid clustering with principal component analysis (PCA) dimensional reduction and Dynamic Time Warping (DTW) distance measurement is utilized to conduct the cluster analysis. Based on the clustering result, an energy schedule-setting matrix is developed with one-hot coding for data-driven energy consumption forecasting. The new schedule setting matrix is compared with the default setting and calendar data in a single-layer neural network (NN) model (data-driven building energy prediction model).

The dataset is clustered into three different time scales, resulting in two clusters for Year–Month data, three clusters for Week–Day data, and two clusters for Day–Hour data. When compared with the default fixed-schedule setting, the schedule setting based on the clustering results improves data-driven predictive models by 25.7%. The data-driven schedule setting also outperforms calendar data, commonly used in data-driven BEM, by 9.2%. Hence, using a schedule setting based on the clustering of building energy profiles may be more suitable for data-driven BEM establishment.

## 2. Methodology

### 2.1. Research Framework

This paper aims to design a schedule-setting matrix based on clustering results of building energy profiles for data-driven BEM in non-residential buildings.

To achieve the objectives, a research framework is proposed. The framework involves historical energy data collection and data processing. Clustering building energy usage patterns in three different time scales (Year–Month, Week–Day, and Day–Hour) to explore energy usage profile patterns. An energy schedule-setting matrix is then developed with one-hot coding for data-driven energy consumption forecasting. The performance of the proposed schedule-setting matrix is further evaluated using building a single-layer neural network model and comparing its accuracy in predicting hourly energy consumption with three types of occupation behaviour data: calendar data, default schedule settings, and clustering results. To facilitate the framework, several key technologies, including data preparation (discretization, segmentation, and cleaning), data representation (PCA dimension reduction), distance measurement (Euclidean distance and DTW distance), clustering algorithm (K-medoids clustering), and data-driven BEM (single-layer neural network) are involved.

Figure 1 shows the framework of this research and step-by-step procedure is described in the following sections.

### 2.2. Data Preparation

The original database is the one-year hourly energy consumption data of 200 office buildings extracted from non-residential buildings in Guangdong, China. The original database is illustrated in Section 3.1 The entire data preparation process included data segmentation for clustering preparation, discretization of building properties data, and data cleaning for missing data process.

Data segmentation was used to process the data into the appropriate dimension according to the time scales. For Year–Month clustering, the data were segmented into 12 dimensions data which was used to demonstrate the change probably due to climate conditions and Heating, Ventilation and Air Conditioning (HVAC) system energy consumption, which often exhibits seasonal features. For Week–Day clustering, the data were segmented into 7 dimensions vector which was used to capture the day's difference in occupancy levels and activity patterns on energy consumption, as these features may depend on whether it is a weekday or weekend/holiday. For Day–Hour clustering, the dataset was processed into 24 dimensions data to represent the hour of the day which was used to capture the features such as occupancy schedules, lighting, and equipment usage which were more likely to repeat in 24 h.

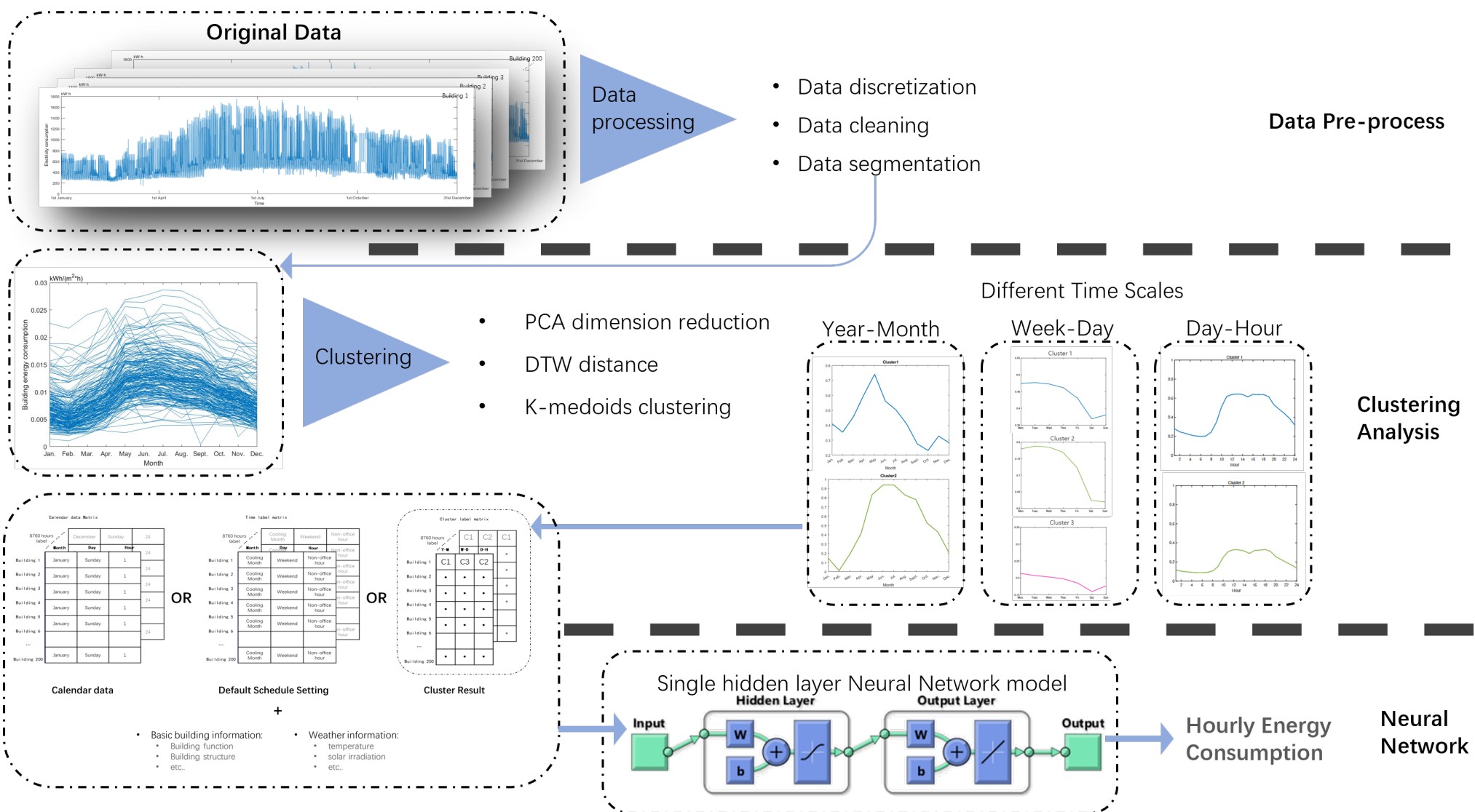

**Figure 1.** Methodology of building electrical energy consumption pattern recognition.

### 2.3. Data Representation

Data representation was one of the most important issues for high-dimensional data clustering. In this study, 12D monthly data for annual analysis, 7D daily data for weekly analysis, and 24D hourly data for daily analysis were investigated. The multidimensional data were difficult for most clustering algorithms to handle. Therefore, PCA was firstly applied to significantly reduce the data dimension while maintaining the character of the original data.

PCA is the most popular dimension reduction method which projects the higher-dimensional $d$-D data to a new lower-dimensional $d'$-D hyperplane that minimizes the projection error. Therefore, its optimization objective is Equation (1)

$$\min_{w} -tr(W^T X X^T W)$$
$$s.t.\ W^T W = I \tag{1}$$

$W$—Transformation matrix;
$X^T X$—Covariance matrix.

Equation (2) can be obtained by applying the Lagrange multiplier to Equation (1).

$$X^T X \mathbf{w_i} = \lambda_i \mathbf{w_i} \tag{2}$$

$\lambda_i$—$\lambda$ is the eigenvalue and $i$ indicated the order of the eigenvalue; c
$\mathbf{w_i}$—The eigenvectors represented using the eigenvalues $\lambda_i$.

For the determination of $d'$ (dimension of low-dimensional space), contribution rate (CR) $t = 95\%$ is used in this study.

$$\frac{\sum_{i=1}^{d'} \lambda_i}{\sum_{i=1}^{d} \lambda_i} \geq t \tag{3}$$

### 2.4. Distance Measure

The distance measure is another key parameter for time-series clustering. Two distance measure indexes were used in this study: Euclidean distance and DTW distance. Euclidean distance was used to measure single-dimension data for k-means clustering, and it was calculated as Equation (4):

$$dist_{Euclidean}(x,y) = \sqrt{\sum |x-y|^2} \tag{4}$$

DTW [23] distance is a typical technique in the addressing time-series related problems [24], and also practical and suitable for the building energy field [25]. In this study, it is used to measure the distance between multi-dimension data for k-medoids clustering, and it is calculated following the Equations (5)–(7) [26].

Using the Euclidean as the distance metric, the $d_{mn}(X,Y)$ which is the distance between the $m$th element in vector $\mathbf{X}$ and the $n$th element in vector $\mathbf{Y}$, is specified in the Equation (5).

$$d_{mn}(\mathbf{X},\mathbf{Y}) = dist_{Euclidean}(x_m, y_n) = \sqrt{\sum |x_m - y_n|^2} \tag{5}$$

Collation of $d_{mn}(\mathbf{X}, \mathbf{Y})$ can form a Distance/Confusion matrix **D** shown in Equation (6):

$$\mathbf{D} = \begin{bmatrix} d_{11} & d_{21} & \cdots & d_{M1} \\ d_{12} & d_{22} & \cdots & d_{M2} \\ & & \ddots & \\ \vdots & \vdots & d_{mn} & \vdots \\ & & & \ddots & \\ d_{1N} & d_{2N} & \cdots & d_{MN} \end{bmatrix} \tag{6}$$

$dist_{DTW}$ aims to minimise the total distance of the path through the distance matrix **D**. The path should start at $d_{11}$, end at $d_{MN}$ and follow the movement rules: · Vertical moves: $(m, n) \rightarrow (m + 1, n)$; · Horizontal moves: $(m, n) \rightarrow (m, n + 1)$; · Diagonal moves: $(m, n) \rightarrow (m + 1, n + 1)$;

$$dist_{DTW}(\mathbf{X}, \mathbf{Y}) = \min \sum_{path} d_{mn}(\mathbf{X}, \mathbf{Y}) \tag{7}$$

### 2.5. Clustering Algorithm

K-medoid clustering is a popular algorithm used in building energy analysis [18] and it was chosen over other partitional-based clustering methods (such as k-means) as the clustering algorithm for three reasons. Firstly, k-medoid clustering can handle non-Euclidean distance measures, whereas k-means clustering assumes that the distance metric is Euclidean. It is essential for this study because both Euclidean distance and DTW distance were used as distance measures and the k-medoid clustering was a better choice to analyse the results. Secondly, k-medoid clustering is less sensitive to the outlines because it uses the actual data point as the cluster medoids and it is less affected by the extreme values than the mean values. Lastly, k-medoid clustering can be more computationally efficient than k-means clustering, especially for large datasets. Therefore, in this study, k-medoid clustering may be more suitable than k-means clustering or other partitional-based clustering methods due to its efficiency, robustness to outliers and ability to handle non-Euclidean distance measures.

It aims to partition $n$ data into $k$ clusters alongside the distance index (Euclidean or DTW distance in this study). Partitioning Around Medoids (PAM) [27] is applied to solve the k-medoids problem, which iterates on the build-steps and the swap-step to achieve the minimized sum distance from medoids to cluster members.

Technically, the algorithm defines the initialization medoid position according to the Arthur and Vassilvitskii method [28]. It is a more intelligent way to determine the initial medoid and benefits the clustering process to avoid local minimization distance and achieve global optimization. Instead of the random selection which is used in the traditional K-means algorithm, the Arthur and Vassilvitskii method uses a probabilistic approach to select the initial centroids that are distant from each other.

Then, the iterative steps were applied to find the best clustering solution: Build-step associates $k$ clusters with a potential medoid. Swap-step tests every point in the cluster as a potential medoid to achieve the minimum sum of within-cluster distances. With the new medoid, $n$ data points were assigned to the clusters with the closest medoids.

### 2.6. Cluster Validation Matrix

In this study, the reference model based on domain knowledge was not set and the choice of the optimal cluster number was based on the elbow method [17] and the Davies–Bouldin Index ($DBI$) [29].

The *DBI* is a common internal cluster performance indicator, and a smaller value indicates better performance. It is calculated using Equation (8).

$$DBI = \frac{1}{k}\sum_{i=1}^{k}\max_{i\neq j}\left(\frac{avg(C_i) + avg(C_j)}{d_{cen}(C_i, C_j)}\right) \tag{8}$$

$k$—Cluster number;
$avg(C)$—The average distance between samples in Cluster $C$;
$d_{cen}(C_i, C_j)$—The distance between the cluster center of Cluster $C_i$ and Cluster $C_j$.

*2.7. Data-Driven BEM*

Artificial Neural Network (ANN) is the most widely used machine learning algorithm and it has been successfully applied to building energy modelling. The previous studies indicated that ANN can achieve an extremely accurate prediction, a Root Mean Square Error (RMSE) of 1.2% [30]. In this study, ANN was established to prove the clustering result can bring an improvement to the prediction accuracy rather than achieve the best result. Therefore, a single-layer neural network was used in this stage.

To set the NN, the initial step was to split the dataset into three subsets: a training set of 70%, a validation set of 15%, and a test set of 15%. The activation function employed was the sigmoid function, while the Levenberg-Marquardt algorithm was used for training. Additionally, the cost function used during training was the mean square error and the maximum epoch number was set as 1000. The accuracy of the ANN model will be evaluated using Mean Squared Error (MSE) (Equation (9)), Mean Absolute Error (MAE) (Equation (10)), Coefficient of Variation (CV) (Equation (11)), and Coefficient of Determination ($R^2$) (Equation (12)).

Given $\hat{Y}$, predict value, and $Y^m$, measured data, the metrics can be calculated as:

$$MSE = \frac{1}{n}\sum_{i=1}^{n}(\hat{Y}_i - Y_i^m)^2 \tag{9}$$

$$MAE = \frac{1}{n}\sum_{i=1}^{n}|\hat{Y}_i - Y_i^m| \tag{10}$$

$$CV = \frac{\sqrt{\sum(\hat{Y} - Y^m)^2}}{\sum Y^m} \tag{11}$$

$$R^2 = 1 - \frac{SS_{res}}{SS_{tot}} = 1 - \frac{\sum_{i=1}^{n}(\hat{Y}_i - Y_i^m)^2}{\sum_{i=1}^{n}(Y_i^m - mean(Y_i^m))^2} \tag{12}$$

## 3. Experiment and Result

*3.1. Database Illustration*

200 non-residential buildings in Guangdong, China were selected as the objectives. They covered four building functions (Office building 112, Commercial building 23, Restaurant and hotel building 21, and Mixed-use building 44) and six building structures (Glass curtain wall structure, reinforced concrete structure, brick and concrete structure, shear wall structure, frame structure, and steel structure). Aiming to better investigate the energy usage pattern and extract knowledge from clustering results, eight building-related parameters were surveyed and updated in the database. They were: (1) The building gross floor area, (2) the air conditioning system, (3) the heating system, (4) the external wall insulation style, (5) the external wall material, (6) the glazing material, (7) the window frame material, and (8) the window type.

In terms of data discretization, numbers were used for ordinal data (such as glazing material) to show the thermal performance ranking of the features, and one-hot encoding was applied to nominal features (such as building function), which had no obvious quantitative relationship with building thermal performance. The detailed information is shown in Table 1.

**Table 1.** Building-related attribute and possible attribute value in the dataset.

| Building-Related Parameters | Properties | Discretization |
| --- | --- | --- |
| Building function | 'Office building', 'Commercial building', 'Restaurant and hotel building', 'Mixed-use building' | One-hot |
| Structure | 'Glass curtain wall', 'Reinforced concrete', 'Brick and concrete', 'Shear wall', 'Frame', 'Steel' | One-hot |
| Building gross area | Scale (minimum 525 m$^2$, maximum 216,000 m$^2$) | - |
| Air conditioning system | 'Centralized air-conditioning system', 'Fan coil and fresh air system', 'Split air conditioning system' | One-hot |
| Heating System | 'Radiator', 'Air conditioner' | One-hot |
| External wall insulation style | 'Internal Insulation', 'External Insulation' | One-hot |
| Exterior wall material | 'Brick', 'Concrete', 'Dinas bricks' | Numbers |
| Glazing Material | 'Clear', 'Coated', 'Low-e' | Numbers |
| Window Framework | 'Aluminum alloy', 'Steel', 'PVC', 'Thermal insulation aluminium profile' | Numbers |
| Window Type | 'Single-glazing','Double-glazing' | Numbers |

For each building, the energy consumption data were recorded for 1 year at 1-h intervals. Building total electricity consumption was recorded in four parts: (1) lighting and plug-in; (2) heating, ventilation and air conditioning; (3) equipment; (4) accessory equipment (i.e., elevator, large laundry equipment); electricity consumption. Electrical energy consumption data, which is the sum of the four energy consumption items was used for clustering.

All the data went through a data-cleaning system to identify the outliers. Missing data in the database often appeared as five or more consecutive values. The data collection step shows that building hourly energy consumption could change trend in 5 h. Therefore, it was unreasonable to impute the missing data with the interpolation method. In this experiment, the missing data are directly deleted rather than further processed. After the data cleaning, for 200 buildings, there were 1,632,067 hourly energy records in total.

For a better understanding of the database, Figure 2 illustrates the hourly energy consumption data for a sample building and Figure 3 shows the average energy consumption level of 200 buildings in the database. The average energy consumption level was measured using electricity consumption (kWh) per area per hour. Different building types are coloured with different colours. The average energy consumption of 200 buildings was 0.010 kWh/(m$^2$·h) and is shown with the dash line in the figure.

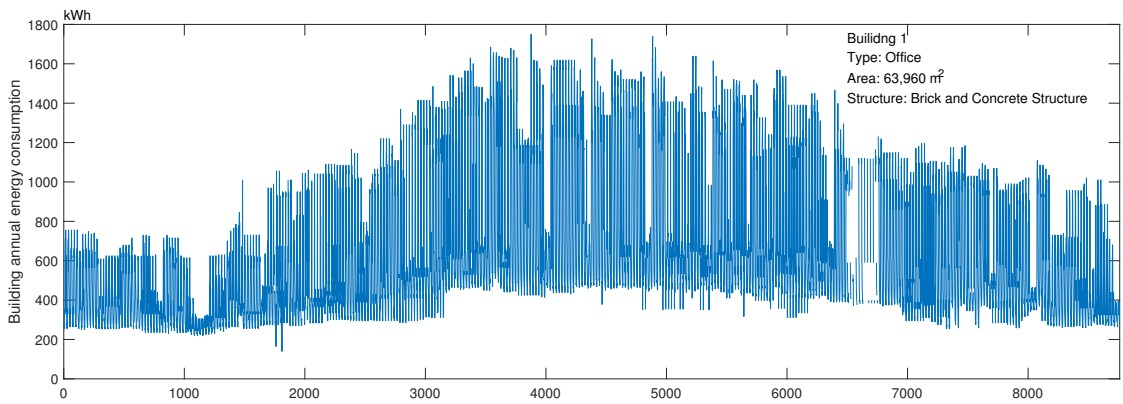

**Figure 2.** Hourly energy consumption data for the sample buildings throughout the year (8760 h).

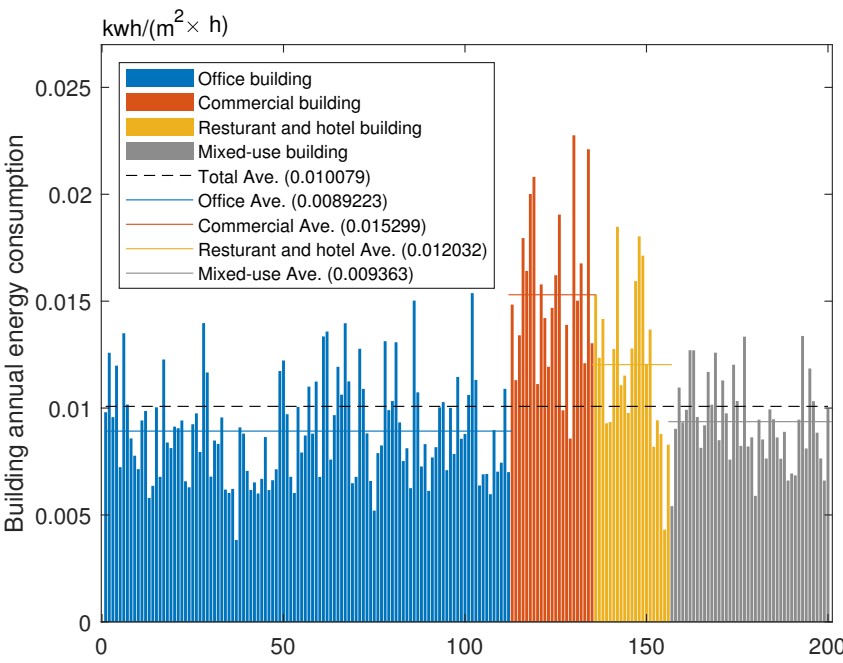

**Figure 3.** Demonstration of the database by building types and energy consumption level.

### 3.2. 1D-Data Clustering Result

K-medoids with Euclidean distance was firstly applied for single-dimension data (statistical mean value) (Annually, monthly, daily, and hourly). The optimal number of clusters was evaluated using DBI and the result shown in Figure 4. The optimal number of clusters and the related best total sum of distance are shown in Table 2.

**Table 2.** Single dimension cluster result.

|  | Annually Figure 4a | Monthly Figure 4b | Daily Figure 4c | Hourly Figure 4d |
|---|---|---|---|---|
| Number of data | 200 | 2400 | 72,671 | 1,632,067 |
| Number of clusters | 8 | 11 | 12 | 2 |
| Sum of distances | $6.00 \times 10^{-5}$ | $7.29 \times 10^{-4}$ | $2.59 \times 10^{-2}$ | 32.41 |

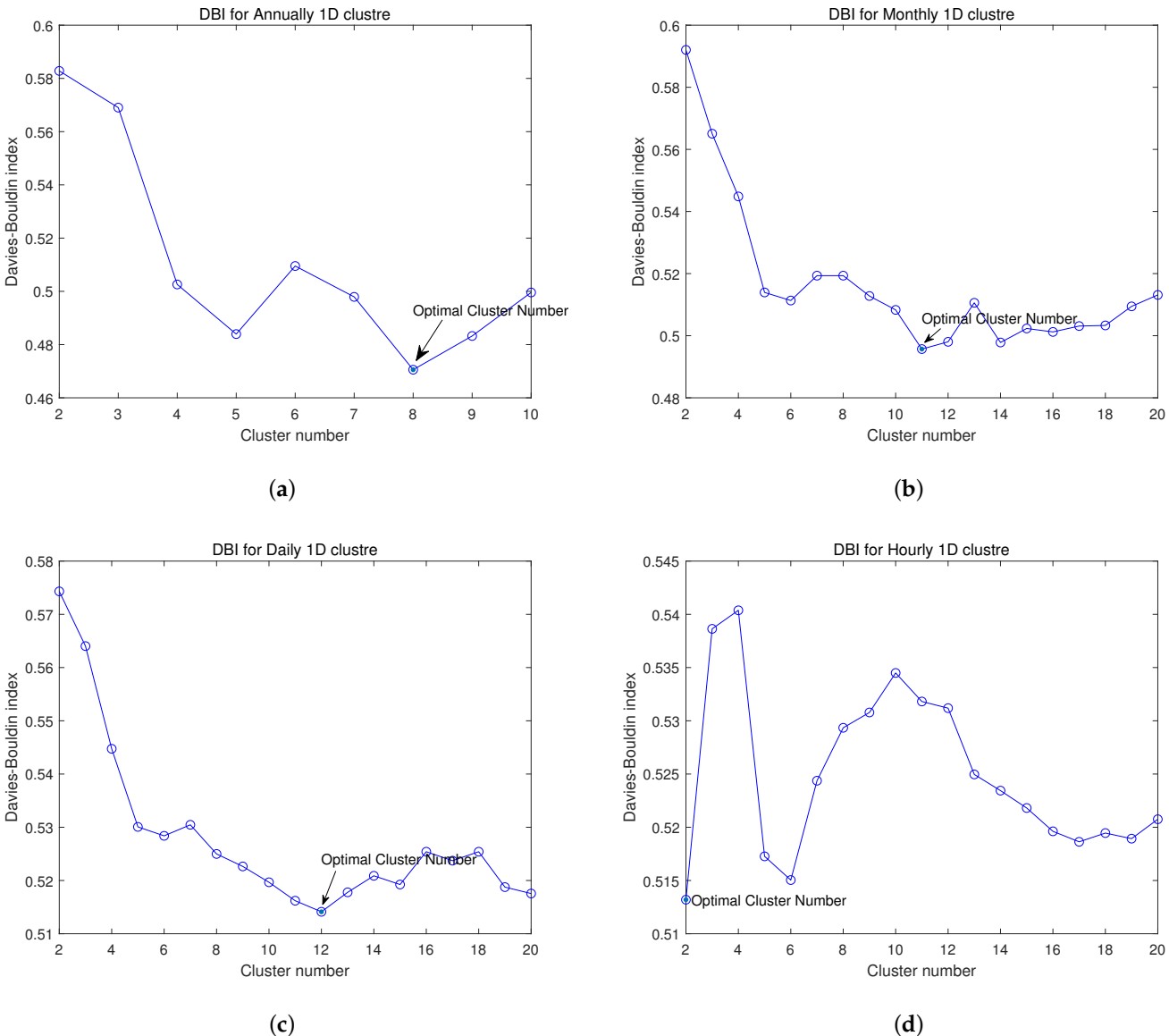

**Figure 4.** Optimized cluster number for 1D data cluster. (**a**) Annually; (**b**) Monthly; (**c**) Daily; (**d**) Hourly.

With the 1D cluster result, Figure 5 visualizes the heatmap in different time scales to analyse the dependence and correlation of recommended schedule setting method of ASHARE on the cluster result. Figure 5 shows the relevance between 1D data cluster result and schedule classification, indicating it is reduced as the time scale becomes shorter. Even with the most relevant monthly clustering and cooling/heating month classification, its *Cramér's V* was only 0.571, which was only a weak correlation (0.5 < *Cramér's V* < 0.7). Therefore, the following study does not set the reference model according to the domain knowledge but uses NN classification for knowledge discovery according to the clustering result.

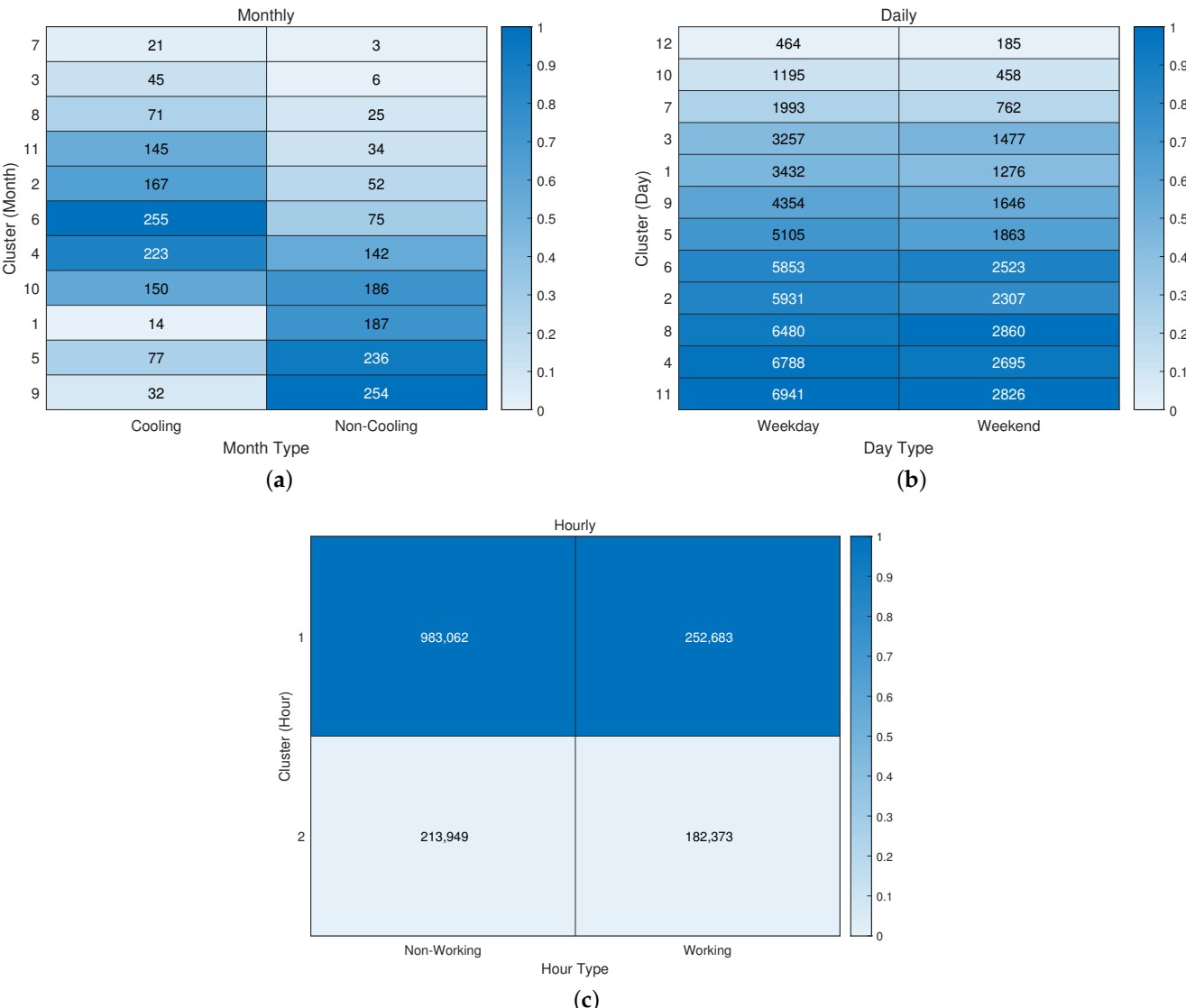

**Figure 5.** Heatmap visualization for 1D cluster result and default schedule setting. (**a**) Monthly Clusters vs. Month Type; (**b**) Daily Clusters vs. Day Type; (**c**) Hourly Clusters vs. Hour Type.

### 3.3. Year–Month Data Clustering Result

For Year–Month clustering, the original data were segmented and averaged as 200 12-dimension time-series data, which is visualized in Figure 6. The monthly energy consumption is the average hourly energy consumption within the month (Units kWh/m$^2$· h).

The original database was firstly normalized according to the maximum and minimum of building monthly energy consumption in the year. Then, PCA was applied to the database. With a 95% CR, the original database has been reconstructed as a 4-Dimension database. Figure 7 visualizes 200 4D data after dimensional reduction.

After PCA, the database was still a multi-dimension database. Therefore, the DTW distance between the 4D data were calculated, and Figure 8 shows the Davies–Bouldin index variation with different numbers of clusters. The cluster number of Year–Month data were selected as 2.

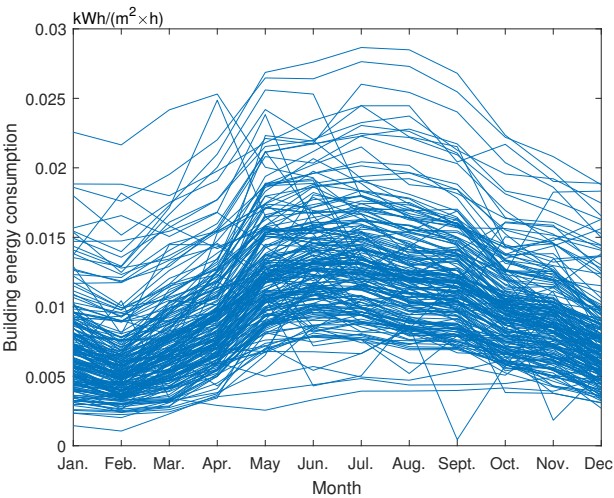

**Figure 6.** 12D visualization of Year–Month data (original data).

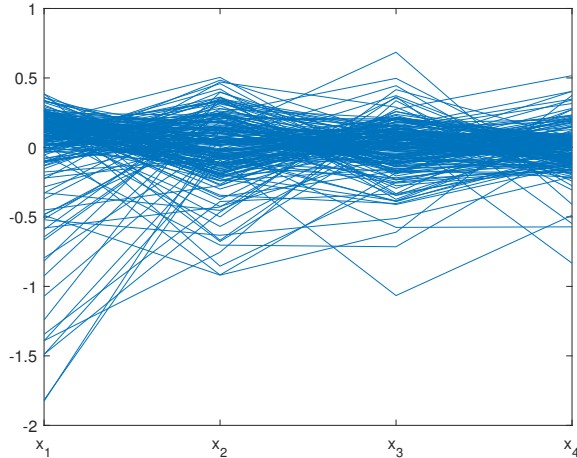

**Figure 7.** 4D visualization of Year–Month data (after PCA with 95% CR).

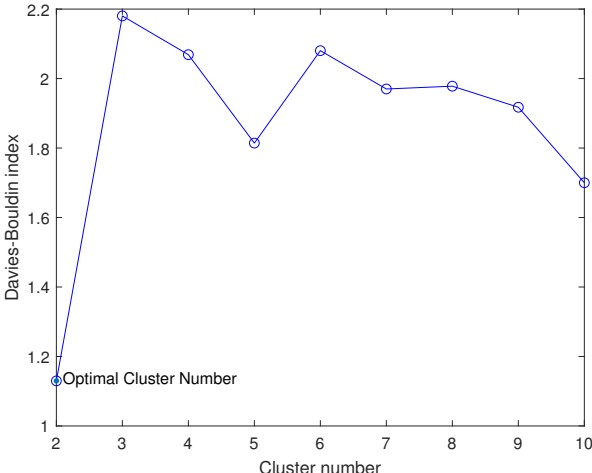

**Figure 8.** Davies–Bouldin index of Year–Month clustering with DTW distance (The optimal cluster number = 2).

Figure 9 shows the clustering results in the 4D version, while Figure 10 shows the clustering results in the 12D version. Figure 11 shows the schematic illustration of two

clusters Year–Month time-series data, which is abstracted from Figure 10. The key changing point analysis is described in Section 4.1.

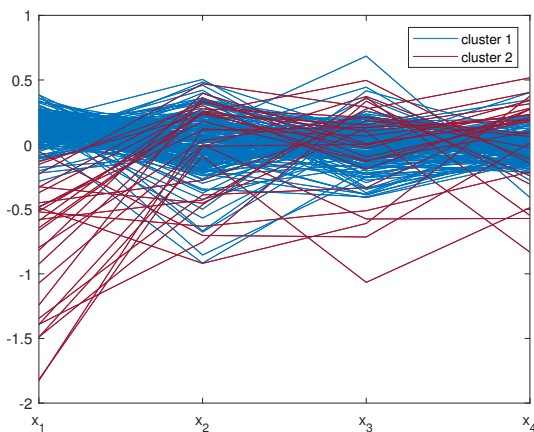

**Figure 9.** Year-Month clustering result in 4D version (cluster number = 2).

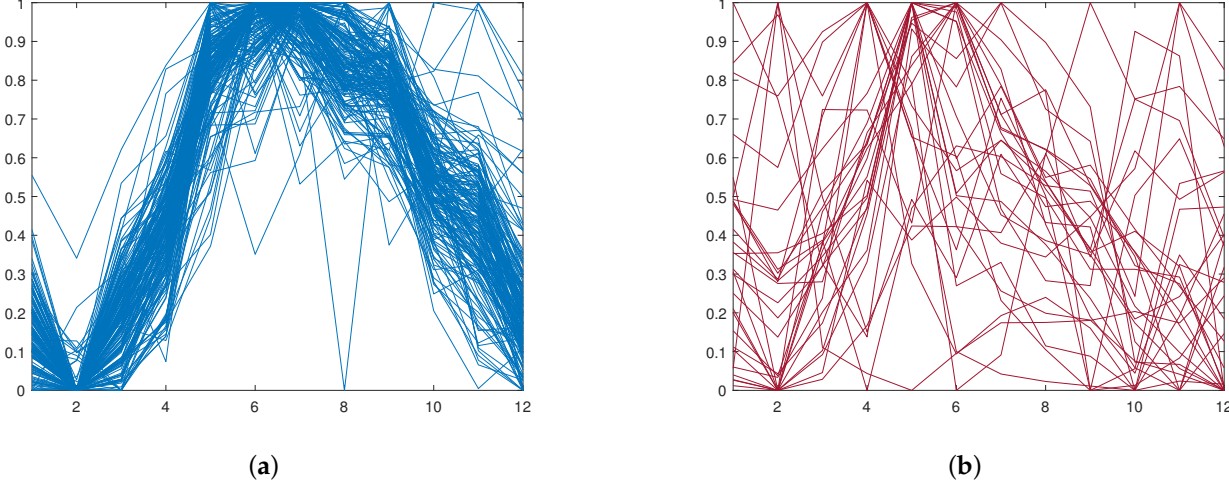

**Figure 10.** Year–Month clustering result in 12D version (cluster number = 2). (**a**) Cluster 1; (**b**) Cluster 2.

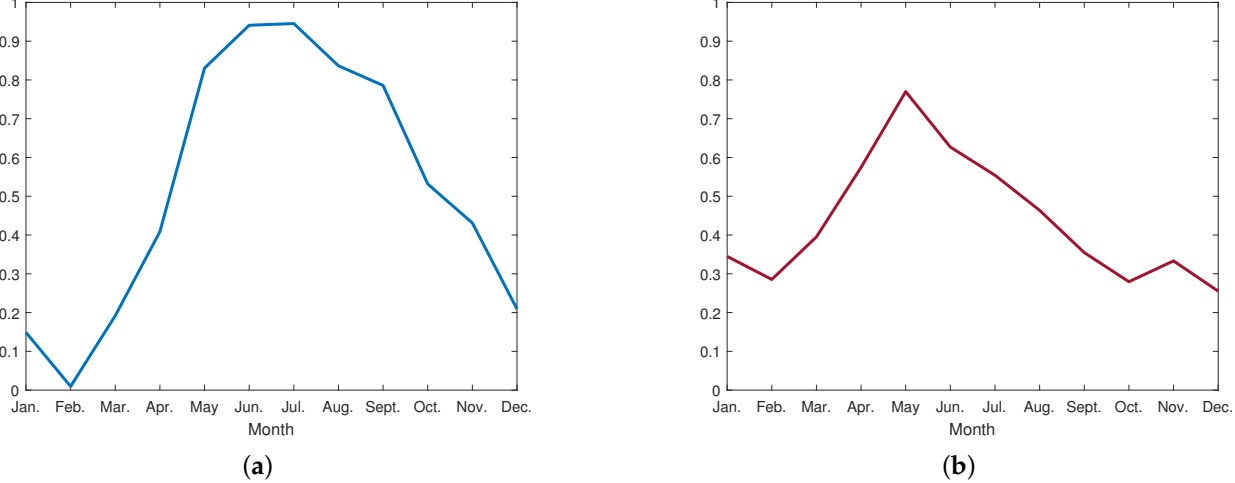

**Figure 11.** Schematic illustration of two clusters Year–Month data. (**a**) Cluster 1; (**b**) Cluster 2.

### 3.4. Week–Day Data Clustering Result

For Week–Day data clustering, the database was again restructured. Parameter 'Daily Energy Consumption' was valued as the daily average energy consumption per square meter per hour. Monday was set as the start of the week and there are 52 complete weekly data (364 days) in the database for each building, which was extracted for Week–Day data cluster analysis. After eliminating missing data, there were 10,258 valid 7D data in the Week–Day database. Figure 12 visualizes 200 Week–Day 7D data (randomly selected) after normalization along with the annual building energy consumption.

Same as the Year–Month database, PCA dimension reduction with a 95% CR was first applied to the database after normalization. The Week–Day database was reconstructed as a single-dimension database. Figure 13 visualizes 10,258 1D data (1st principle component) in the Week–Day database. Then, Figure 14 shows the Davies–Bouldin index (with Euclidean distance) variation with different numbers of clusters. The optimal cluster number of Week–Day data was 3.

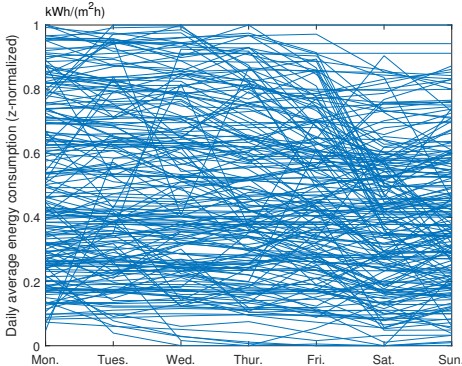

**Figure 12.** 7D visualization of Week–Day data (after normalization).

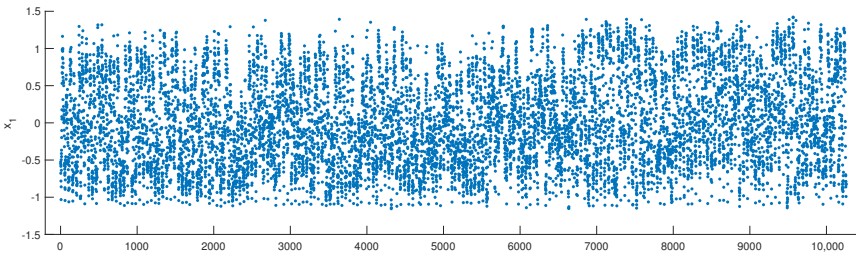

**Figure 13.** 1D visualization of Week–Day data (after PCA with 95% CR).

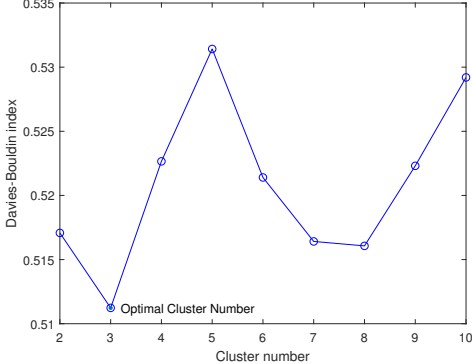

**Figure 14.** Davies–Bouldin index of Week–Day clustering with Euclidean distance (The optimal cluster number = 3).

The Week–Day clustering results with three clusters are shown in Figures 15–17 in different versions.

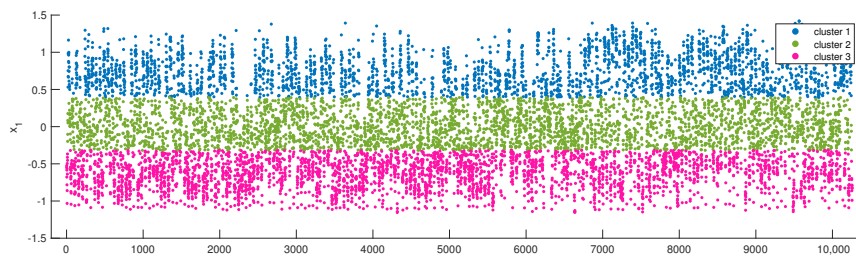

**Figure 15.** Week-Day clustering result in 1D version (cluster number = 3).

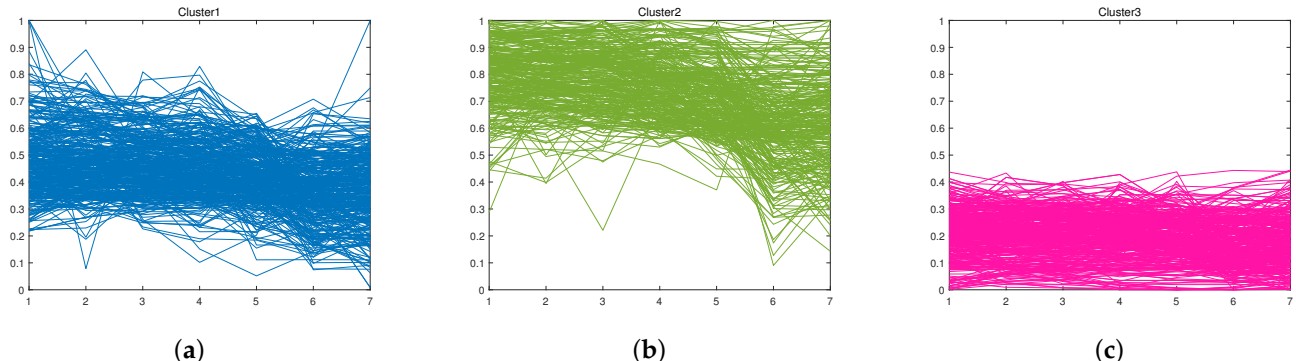

**Figure 16.** Week-Day clustering result in 7D version (cluster number = 3). (**a**) Cluster 1; (**b**) Cluster 2; (**c**) Cluster 3.

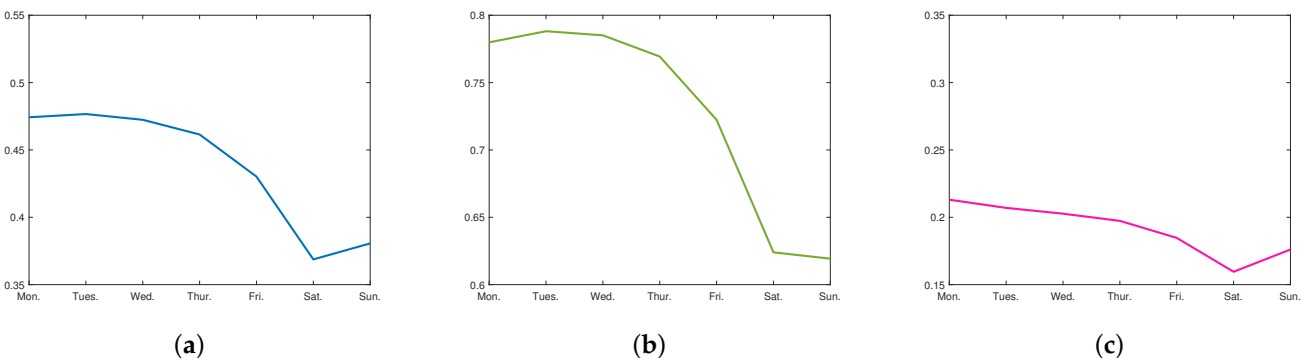

**Figure 17.** Schematic illustration of three clusters Week–Day data. (**a**) Cluster 1; (**b**) Cluster 2; (**c**) Cluster 3.

### 3.5. Day–Hour Data Clustering Result

The original database was firstly reorganized into an array of $24 \times 365 \times 200$. For each building, the data were normalized to the maximum and minimum hourly energy consumption. Figure 18 visualizes 365 normalized 24-dimensional data for a building (randomly selected). After data cleaning, there were 52085 days with valid data left in the dataset for clustering.

After PCA dimension reduction with a 95% CR, the 24D Day–Hour data has two principal components. Figure 19 visualizes the top two principle components in the Day–Hour dataset after dimension reduction. According to the methodology, DTW distance was calculated, and Figure 20 shows the Davies–Bouldin index variation with different numbers of clusters. The cluster number of Day–Hour data was selected as 2.

The Day–Hour clustering results with two clusters are shown in Figures 21 and 22.

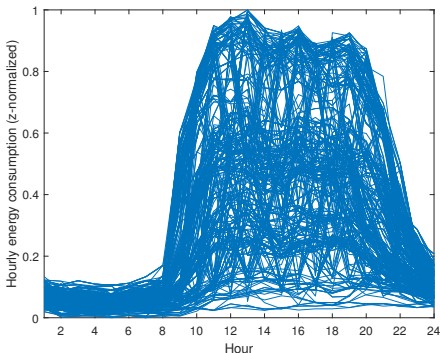

**Figure 18.** 24D visualization of Day–Hour data (after z-normalization).

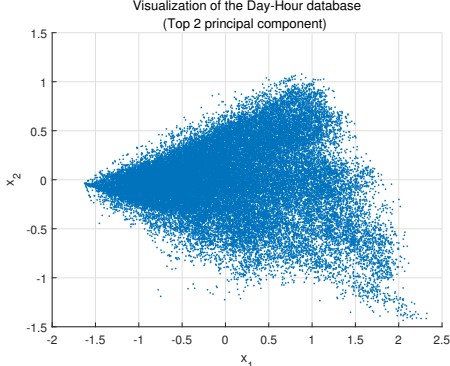

**Figure 19.** 2D visualization of Day–Hour data (Top two principle component).

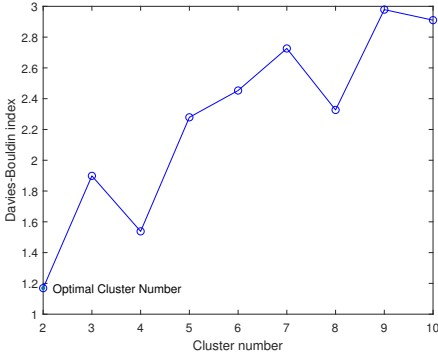

**Figure 20.** Davies–Bouldin index of Day–Hour clustering with DTW distance (The optimal cluster number = 2).

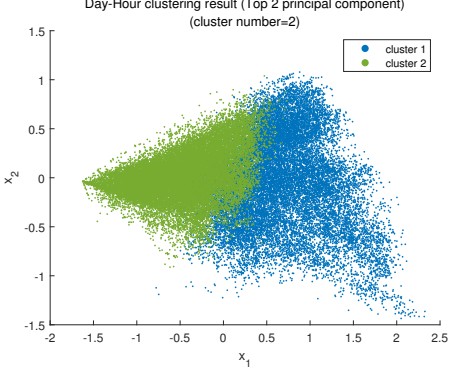

**Figure 21.** Day–Hour clustering result in 2D version (cluster number = 2).

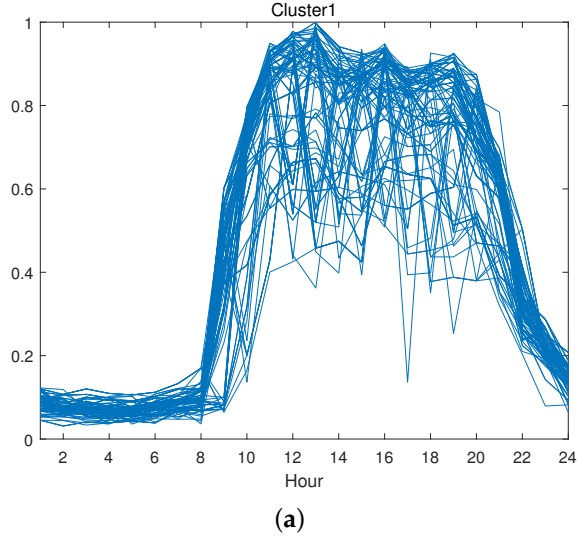

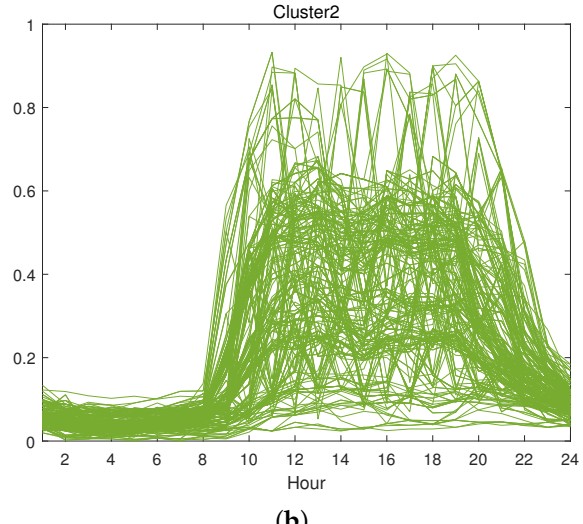

(**a**)

(**b**)

**Figure 22.** Day-Hour clustering result in 24D version (cluster number = 2). (**a**) Cluster 1; (**b**) Cluster 2.

## 4. Discussion

### 4.1. Clustering Result Analysis and Knowledge Discovery

#### 4.1.1. Year–Month Cluster

The Year–Month data were partitioned into two clusters and the schematic is illustrated in Figure 11 (detailed information listed in Table 3).

**Table 3.** Year–Month cluster description.

| Cluster | Coordinate | Average | Maximum | Minimum | Proportion |
|---|---|---|---|---|---|
| Y-M-1 | [0.15, 0.01, 0.19, 0.41, 0.83, 0.94, 0.95, 0.84, 0.79, 0.53, 0.43, 0.21] | 0.52 | 0.95 (July) | 0.01 (February) | 85.5% |
| Y-M-2 | [0.34, 0.29, 0.40, 0.57, 0.77, 0.63, 0.55, 0.46, 0.35, 0.28, 0.33, 0.26] | 0.44 | 0.77 (May) | 0.26 (December) | 14.5% |

Cluster Y-M-1 is the common Year–Month energy profile, accounting for 85.5% of total data. In this cluster, the high energy consumption lasts for several months and the energy consumption change throughout the year is mild. It could be described as 'trapezium-type' without a sharp rise or drop. The highest point usually appears in June or July while the lowest point was located in February.

Correspondingly, Cluster Y-M-2 was rare (14.15%) and was more similar to the 'triangle-type'. It has one obvious and sharp peak, and the peak point could appear in any month from April to September. However, in January-March and October-December, the energy consumption value remained stable and low.

#### 4.1.2. Week–Day Cluster

The Week–Day data were partitioned into three clusters and the schematic is illustrated in Figure 17 (detailed information listed in Table 4).

**Table 4.** Week–Day cluster description.

| Cluster | Coordinate | Average | Weekdays | Weekends | Deviation * | Proportion |
|---|---|---|---|---|---|---|
| W-D-1 | [0.47, 0.48, 0.47, 0.46, 0.43, 0.37, 0.38] | 0.437 | 0.462 | 0.375 | 0.087 | 36.0% |
| W-D-2 | [0.78, 0.79, 0.79, 0.77, 0.72, 0.62, 0.62] | 0.727 | 0.770 | 0.620 | 0.15 | 28.8% |
| W-D-3 | [0.21, 0.21, 0.20, 0.20, 0.18, 0.16, 0.18] | 0.191 | 0.200 | 0.170 | 0.03 | 35.2% |

* Deviation between mean value of weekday and weekend.

Different from Year–Month clustering results, there was no domain type in W-D clustering, and the three clusters accounted for approximately equal proportions. These three types could be easily distinguished using weekly average energy consumption and they were similar in shape. However, the deviation between weekdays and weekends decreased as the average energy consumption for the whole week decreased. In the W-D-1 and W-D-2, there was an obvious difference between weekdays and weekends with Friday as a transition. Compared to the other two cluster, W-D-3 could be considered as a horizontal curve with lower load.

### 4.1.3. Day–Hour Cluster

The Day–Hour data were partitioned into two clusters and the schematic is illustrated in the Figure 23 (detailed information listed in Table 5).

**Table 5.** Day–Hour cluster description.

| Cluster | Coordinate | Average | Key Changing Point | Proportion |
|---|---|---|---|---|
| D-H-1 | [0.28, 0.25, 0.23, 0.22, 0.20, 0.20, 0.20, 0.24, 0.35, 0.51, 0.62, 0.64, 0.64, 0.64, 0.61, 0.64, 0.64, 0.64, 0.62, 0.53, 0.48, 0.44, 0.34, 0.32] | 0.439 | [7, 12, 15, 18] | 42.0% |
| D-H-2 | [0.12, 0.10, 0.10, 0.09, 0.09, 0.09, 0.09, 0.11, 0.15, 0.25, 0.31, 0.33, 0.33, 0.31, 0.29, 0.32, 0.32, 0.33, 0.31, 0.26, 0.23, 0.20, 0.17, 0.13] | 0.209 | [7, 12, 15, 18] | 58.0% |

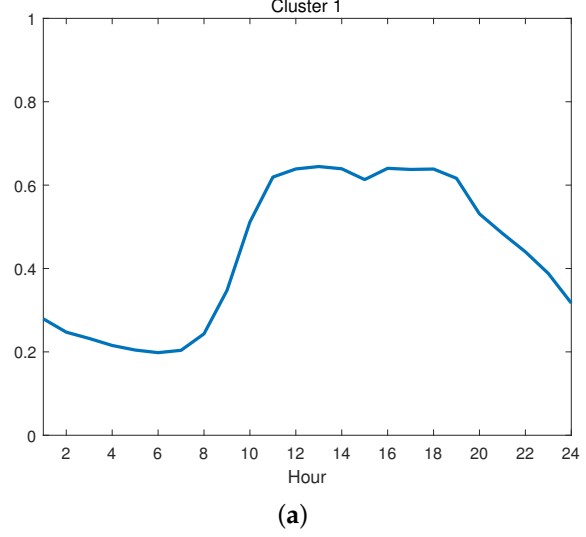

(**a**)

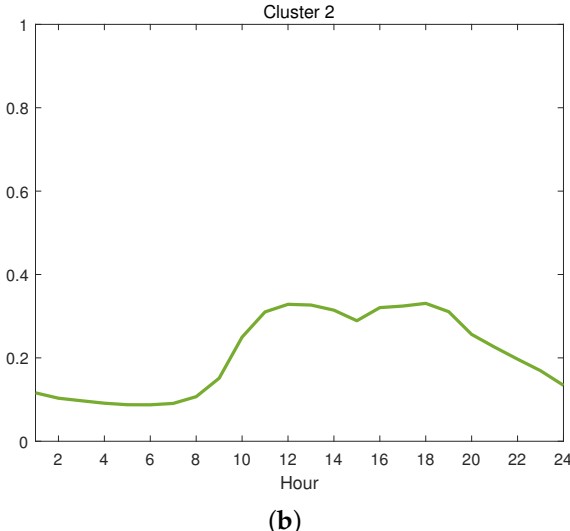

(**b**)

**Figure 23.** Schematic illustration of two clusters Day–Hour data. (**a**) Cluster 1; (**b**) Cluster 2.

In the D–H cluster, although the energy consumption level gap between two clusters was large, it was not difficult to find that they have a similar shape. Specifically, two clusters have the same key changing points at Hour 7, Hour 12, Hour 15 and Hour 18 (change in the sign of the derivative), these key changing points also show changes to the working schedule.

On the other hand, the proportion of the two clusters was very close to the proportion of days with cooling demand in Guangdong. It is reasonable to assume that the Day–Hour cluster not only represents the working schedule, but also has a significance on cooling demand.

For better illustration, Figure 24 shows the heatmap between the month of the year and Day–Hour clusters with the colour scale to the cluster. Table 6 shows the percentage of each cluster in each month's data. According to the result in Table 6, a potential month schedule setting method is given in Table 7.

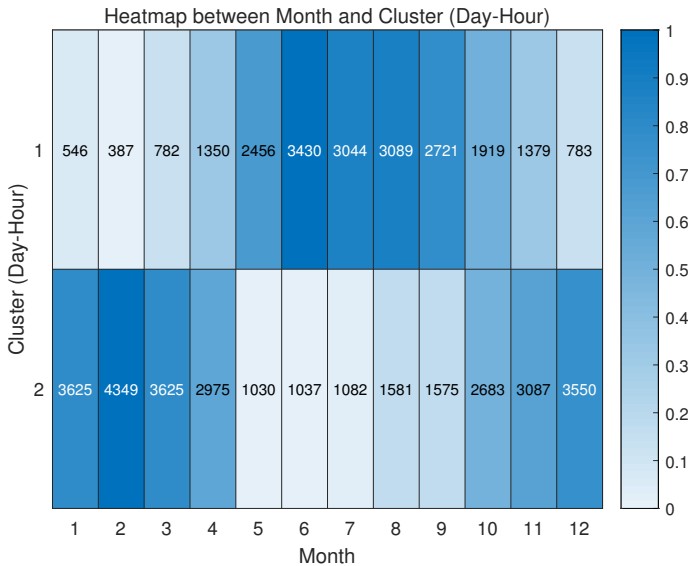

**Figure 24.** Heatmap between Month and Day–Hour Cluster.

**Table 6.** Percentage of cluster in month (%).

|  | January | February | March | April | May | June | July | August | September | October | November | December |
|---|---|---|---|---|---|---|---|---|---|---|---|---|
| **Cluster 1** | 13.09 | 08.17 | 17.74 | 31.21 | 70.45 | 76.78 | 73.77 | 66.14 | 63.33 | 41.69 | 30.87 | 18.07 |
| **Cluster 2** | 86.90 | 91.82 | 82.25 | 68.78 | 29.54 | 23.21 | 26.22 | 33.85 | 36.66 | 58.30 | 69.12 | 81.92 |

**Table 7.** Potential month schedule setting method.

| Category | Month in the Category | Criterion |
|---|---|---|
| Month with high cooling demand | May, June, July, August, September, | Cluster 1 accounts for more than 70% |
| Month with partial cooling demand | April, October, November, | No Cluster accounts for more than 70% |
| Month without cooling demand | January, February, March, December | Cluster 2 accounts for more than 70% |

### 4.2. ANN Model with/without Clustering Result

To test the performance of clustering results on a data-driven prediction model, a single-layer NN model was established and the basic neural network structure is shown in Figure 25.

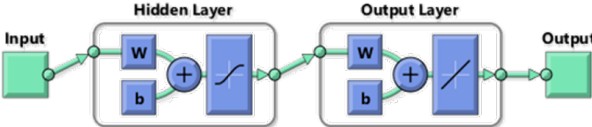

**Figure 25.** Single-layer NN model structure.

The input parameters of the baseline model are composed of three parts:
- Three sets of comparative data (Calendar data, Default Schedule Setting, and Cluster Result);
- basic building information after discretization: Building function, building structure, etc.
- Weather information: Temperature, solar irradiation, etc. (from Guangdong weather files).

The illustration of the input dataset is shown in Figure 26.

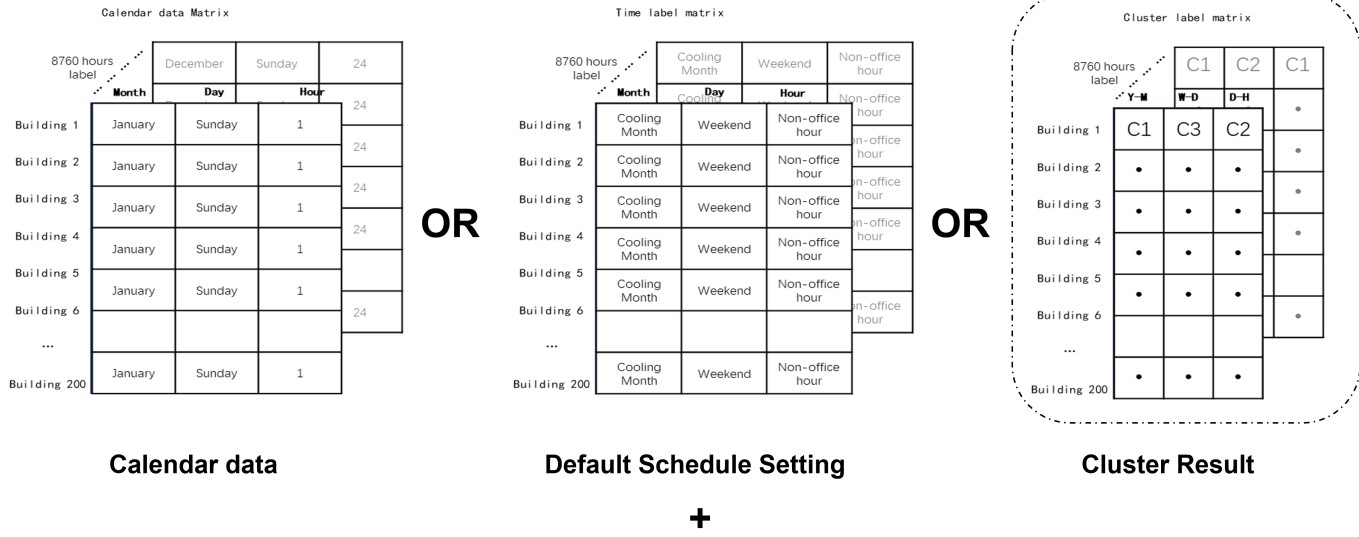

**Figure 26.** Input parameters of NN model.

The first part of the input parameters is three sets of comparative data. The calendar data were taken with numerical values 1–12 representing the month of the year, 1–7 for the day of the week, and 1–24 for the hour of the day. Both default schedule setting and cluster results take the one-hot coding method to represent categories. The second part was a dataset of necessary parameterized building features. The third part was weather information which had a significant influence on building energy consumption.

The number of units in the hidden layer changed from 10 to 30, and the output was hourly energy consumption. The performance of the ANN model was evaluated using MSE, MAE, CV, and R and the result is shown in Table 8.

**Table 8.** NN prediction performance.

| Units in Hidden Layer | Metrics | Calendar Data 1 | Default Schedule 2 | Cluster Result 3 | Improvement 3 from 1 | Improvement 3 from 2 |
|---|---|---|---|---|---|---|
| 10 | MSE | $2.621 \times 10^{-5}$ | $3.560 \times 10^{-5}$ | $2.115 \times 10^{-5}$ | 10.3% | 22.9% |
|  | MAE | $3.627 \times 10^{-3}$ | $4.263 \times 10^{-3}$ | $3.221 \times 10^{-3}$ |  |  |
|  | CV(RMSE) | $4.002 \times 10^{-4}$ | $4.660 \times 10^{-4}$ | $3.591 \times 10^{-4}$ |  |  |
|  | R | 0.777 | 0.680 | 0.825 |  |  |
| 20 | MSE | $2.323 \times 10^{-5}$ | $3.251 \times 10^{-5}$ | $1.852 \times 10^{-5}$ | 10.6% | 24.3% |
|  | MAE | $3.317 \times 10^{-3}$ | $4.014 \times 10^{-3}$ | $2.940 \times 10^{-3}$ |  |  |
|  | CV(RMSE) | $3.767 \times 10^{-4}$ | $4.451 \times 10^{-4}$ | $3.368 \times 10^{-4}$ |  |  |
|  | R | 0.806 | 0.713 | 0.849 |  |  |
| 30 | MSE | $2.072 \times 10^{-5}$ | $3.097 \times 10^{-5}$ | $1.709 \times 10^{-5}$ | 9.2% | 25.7% |
|  | MAE | $3.095 \times 10^{-3}$ | $3.896 \times 10^{-3}$ | $2.761 \times 10^{-3}$ |  |  |
|  | CV(RMSE) | $3.556 \times 10^{-4}$ | $4.349 \times 10^{-4}$ | $3.230 \times 10^{-4}$ |  |  |
|  | R | 0.829 | 0.729 | 0.861 |  |  |

The Improvement in the table is calculated based on CV.

From the Table 8, the key findings could be listed as:

- As the number of hidden units in the hidden layer increased, all three models performed better.
- With the same hidden units in the layer, the clustering result performed better.

- When hidden units was 30, the clustering result had a 25.7% improvement from the default schedule and 9.2% from the calendar data.

## 5. Conclusions

This study applied a clustering algorithm to identify different energy profile patterns for non-residential buildings; and further, based on the clustering results, a schedule-setting method for data-driven building energy models has been proposed.

The study first analysed a comprehensive and high-resolution dataset consisting of one-year historical electricity data from 200 non-residential buildings of different types, over three different time scales collected from Guangdong, China. Key technologies such as PCA dimension reduction, DTW distance, and K-medoid clustering were employed to develop the clustering algorithm.

The clustering result shows that:

- The Year–Month data were partitioned into two clusters: 'Trapezium-type' and 'Triangle-type'.
- The Week–Day data were partitioned into three clusters: 'High Energy Demand Week', 'Medium Energy Demand Week', and 'Low Energy Demand Week';
- The Day–Hour data were partitioned into two clusters: 'Cooling Mode' and 'Non-Cooling Mode'.

A single-layer neural network model was established to test the performance of the clustering results on data-driven energy prediction.

The NN model performance shows that:

- On data-driven predictive models, the clustering-based schedule setting performs much better than the default fixed schedule set which is routinely used in the engineering BEM, resulting in a 25.7% improvement.
- Furthermore, the data-driven schedule setting is more advantageous compared to the calendar data which is used in the data-driven BEM in place with schedule setting, with a 9.2% improvement.

To review the research, the conclusion can be stated.

- K-medoid clustering with DTW distance and PCA dimension reduction could be employed to clustering analysis of building energy profiles and develop a data-related energy schedule setting matrix.
- The proposed data-related schedule matrix performs better when integrated with a data-driven BEM than the default fixed schedule set and calendar data.
- The data-driven BEM could benefit from an appropriate data-related schedule-setting method which would be more flexible and elaborate.

When data are available, future research could be done to explore the potential of the applicability of this approach across different locations and climate conditions. Additionally, with the development of the algorithm, more elaborate algorithms can be applied to this problem to establish an energy-related schedule which is more suitable for data-driven BEM.

**Author Contributions:** Conceptualization, Y.C., Z.Z. and X.Z.; methodology, Y.C., Z.Z. and X.Z.; software, Y.C.; validation, Y.C. and Z.L.; formal analysis, Y.C.; investigation, Y.C.; data curation, Y.C. and Z.L.; writing-original draft preparation, Y.C.; writing-review and editing, Y.C., Z.Z. and Z.L.; visualization, Y.C.; supervision, Z.Z. and X.Z.; All authors have read and agreed to the published version of the manuscript.

**Funding:** This research was funded by the UK Newton Fund and Guangdong Department of Science and Technology OF FUNDER grant number 101005-586174.

**Institutional Review Board Statement:** Not applicable.

**Informed Consent Statement:** Not applicable.

**Data Availability Statement:** Not applicable.

**Conflicts of Interest:** The authors declare no conflict of interest.

**Abbreviations**

The following abbreviations are used in this manuscript:

| | |
|---|---|
| BEM | Building energy model/ modelling |
| PCA | Principal Component Analysis |
| DTW | Dynamic Time Warping |
| NN | Neural Network |
| ASHRAE | American Society of Heating, Refrigerating, and Air-Conditioning Engineers |
| MNLR | Multiple non-linear regression |
| DNN | Deep neural network |
| PAM | Partitioning Around Medoids |
| DBI | Davies–Bouldin Index |
| CR | Contribution Rate |
| ANN | Artificial Neural Network |
| RMSE | Root Mean Square Error |
| MSE | Mean Square Error |
| MAE | Mean Absolute Error |
| CV | Coefficient of Variation |

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
