# Peer review of "Energy Schedule Setting Based on Clustering Algorithm and Pattern Recognition for Non-Residential Buildings Electricity Energy Consumption"

_sustainability, doi:10.3390/su15118750_

Round 1
Reviewer 1 Report
1) English language does not reach the minimum standard.
2) Abstract requires improvement and needs to clearly state the novelty, contribution, and results of the study.
2) The novelty is not clear. If the novelty is in the proposed modified version of the algorithm, its performance must be assessed not only against other algorithms but also against its basic version.
3) It is not clear whether the same method is applicable in other contexts.
4) The motivations for this work are insufficiently explained.
5) The methodology is insufficiently described.
6) A comparison with another method would be interesting
7) Conclusions are not precise, improve give point-wise and add follow-up research area. It repeats the earlier statements
-----
Reviewer 2 Report
The paper presents a data-driven approach to BEM, which combines cluster analysis with PCA to partition a building energy historical dataset into a set of time scales and explore energy usage profile patterns. The resulting schedule-setting matrix is then compared with default settings and calendar data using a one-hidden layer (shallow) neural network. The authors show that the obtained results perform significantly better than the conventional fixed schedule setting.
I think that the paper is interesting and easy to follow. However, some parts and design choices require further clarification. For example, the paper could benefit from a more detailed explanation of the following points:
1) The rationale behind the choice of k-medoid clustering over other partitional-based clustering methods such as the minimum sum-of-square clustering (a.k.a. k-means clustering) which has been widely used in BEM. The results of k-medoids are highly dependent on the initial cluster centers, and solving the problem heuristically can be misleading. While the k-medoids algorithm does not guarantee global optimality, just like the classical k-means algorithm, there are solution approaches that can find and certify the globally optimal clustering solution for large datasets in a reasonable amount of time, see:
- V. Piccialli, A. M. Sudoso and A. Wiegele (2022). SOS-SDP: An Exact Solver for Minimum Sum-of-Squares Clustering, INFORMS Journal on Computing, 34(4):2144-2162. https://doi.org/10.1287/ijoc.2022.1166.
Could the authors discuss the impact of solving the clustering problem for BEM heuristically instead of using an exact approach? It would be interesting for future research to mention this important issue and compare the performance of exact solvers for clustering with heuristic methods.
2) The specific features or variables that were used to partition the building energy historical dataset into three different time scales (Year-Month, Week-Day, and Day-Hour). What were the criteria for selecting these time scales, and how do they relate to the energy consumption patterns in the building?
3) The implementation details of the single-layer neural network model used to compare the data-driven schedule-setting matrix with default settings and calendar data. In order to ensure reproducibility, the authors should describe in more detail the experimental setting of the network, including the number of epochs, regularization, how many hidden layers, etc...
Minor editing of English language required
Author Response
Dear Reviewer,
Please kindly find the attachment which provides a point-by-point response to your comments. We hope our revised manuscript can be accepted for publication.
Best regards,
Yu Cui, Zishang Zhu, Xudong Zhao, and Zhaomeng Li

Round 2
Reviewer 1 Report
----
----
Author Response
Dear Reviewer,
We would like to thank you for your careful reading and helpful comments. A native colleague has polished the English for the manuscript. We hope our revised manuscript can be accepted for publication.
Sincerely,
Yu Cui, Zishang Zhu, Xudong Zhao, Zhaomeng Li
Reviewer 2 Report
The manuscript showed improvement after the authors addressed most of the comments from the reviewer, but they did not address the first comment regarding the significance and implications of utilizing a heuristic approach to solve the clustering problem for BEM, as opposed to the suggested global optimization algorithm (SOS-SDP). It would be beneficial to add a discussion regarding this crucial issue.
Minor editing of English is required.
Author Response
Dear Reviewer,
We would like to thank you again for your careful reading, helpful comments, and constructive suggestions. The corrections in the manuscript and the point-by-point response to the Reviewer's comments are listed in the attachment. We hope our revised manuscript can be accepted for publication.
Sincerely,
Yu Cui, Zishang Zhu, Xudong Zhao and Zhaomeng Li.

Round 3
Reviewer 1 Report
------------
------
Reviewer 2 Report
The authors addressed all the reviewer's concerns. Thanks.